# Anisotropy-driven quantum criticality in an intermediate valence system

Mihael S. Grbić [1,2✉], Eoin C. T. O'Farrell[1✉], Yosuke Matsumoto[1], Kentaro Kuga [1], Manuel Brando[3], Robert Küchler[3], Andriy H. Nevidomskyy [4], Makoto Yoshida[1], Toshiro Sakakibara[1], Yohei Kono[1], Yasuyuki Shimura[1], Michael L. Sutherland [5], Masashi Takigawa[1✉] & Satoru Nakatsuji[1,6,7,8,9✉]

Intermetallic compounds containing *f*-electron elements have been prototypical materials for investigating strong electron correlations and quantum criticality (QC). Their heavy fermion ground state evoked by the magnetic *f*-electrons is susceptible to the onset of quantum phases, such as magnetism or superconductivity, due to the enhanced effective mass ($m^*$) and a corresponding decrease of the Fermi temperature. However, the presence of *f*-electron valence fluctuations to a non-magnetic state is regarded an anathema to QC, as it usually generates a paramagnetic Fermi-liquid state with quasiparticles of moderate $m^*$. Such systems are typically isotropic, with a characteristic energy scale $T_O$ of the order of hundreds of kelvins that require large magnetic fields or pressures to promote a valence or magnetic instability. Here we show the discovery of a quantum critical behaviour and a Lifshitz transition under low magnetic field in an intermediate valence compound $\alpha$-YbAlB$_4$. The QC origin is attributed to the anisotropic hybridization between the conduction and localized *f*-electrons. These findings suggest a new route to bypass the large valence energy scale in developing the QC.

[1] Institute for Solid State Physics (ISSP), University of Tokyo, Kashiwa 277-8581, Japan. [2] Department of Physics, Faculty of Science, University of Zagreb, Bijenička 32, Zagreb HR 10000, Croatia. [3] Max Planck Institute for Chemical Physics of Solids, Nöthnitzer Strasse 40, D-01187 Dresden, Germany. [4] Department of Physics and Astronomy, Rice University, Houston, TX 77005, USA. [5] Cavendish Laboratory, University of Cambridge, J.J. Thomson Avenue, CB3 0HE Cambridge, United Kingdom. [6] Department of Physics, The University of Tokyo, Tokyo 113-0033, Japan. [7] CREST, Japan Science and Technology Agency (JST), Honcho Kawaguchi, Saitama 332-0012, Japan. [8] Institute for Quantum Matter and Department of Physics and Astronomy, Johns Hopkins University, Baltimore, MD 21218, USA. [9] Trans-scale Quantum Science Institute, University of Tokyo, Tokyo 113-0033, Japan. ✉email: mgrbic@phy.hr; eoin.ofarrell@gmail.com; takigawa.masashi@gmail.com; satoru@issp.u-tokyo.ac.jp

A quantum critical point (QCP) occurs when the ground state of a system is continuously tuned between two states[1,2]. The strong incipient quantum fluctuations modify the system's electronic state over large regions of its phase diagram. This has led to the notion that understanding quantum criticality (QC) is the key to understanding emergent phases in materials such as spin liquids and the high temperature superconductors.

Heavy fermion (HF) materials, mainly $f$-electron based intermetallics containing Ce or Yb, have been prototypical for the investigation of QC[1–4]: the enhanced entropy of the Fermi sea makes them susceptible to low temperature instabilities such as magnetism and superconductivity. The energy differences between these possible ground states are small, and therefore can typically be tuned by applying small magnetic field or pressure.

However, the conventional paradigm of QC in HF materials, the Doniach phase diagram and its recent extensions[5], requires a stable valence of the magnetic ion, which for Yb is the 3+ state. In Yb intermetallics with fluctuating valence, such as $YbAl_3$ or $YbAgCu_4$, a large valence fluctuation energy scale $T_0$ makes new phases difficult to achieve; $YbAl_3$ is not known to order magnetically at all[6], while $YbAgCu_4$ requires extremely high pressure[7].

The recently synthesized $YbAlB_4$ is therefore remarkable because QC coexists with intermediate valence, +2.73 and +2.75 for both, $\alpha$ and $\beta$ polymorphs[8], respectively. Both materials have a valence fluctuation scale $T_0 \approx 200$ K[8,9] and a negligible change of valence[10] in magnetic fields up to 40 T. The two polymorphs of $YbAlB_4$ are locally isostructural, with a highly anisotropic

structure, atypical for intermediate valence systems, where sheets of boron separate the layers containing Yb and Al (Fig. 1a). Below the temperature $T_0$ an expected Fermi liquid (FL) ground state should lead to a constant susceptibility ($\chi$) and specific heat coefficient $C_e/T = \gamma$. Unusually, the Ising anisotropy along the $c$ axis dominates[9] in $\chi$, and only for $T < T^* \approx 8$ K it saturates in $\alpha$-$YbAlB_4$, while in $\beta$-$YbAlB_4$ it diverges due to QC at ambient conditions[11]. Nevertheless, the FL ground state of $\alpha$-$YbAlB_4$ is shown by the saturated $\gamma$ value and $T^2$-resistivity ($\rho$) at low temperatures. However, $\rho$ remains very anisotropic[9] with $\rho_{ab}/\rho_c \approx 11$, suggesting that the hybridization is much stronger within the $ab$ plane than along the $c$ axis. In this article we focus on $\alpha$-$YbAlB_4$ and show the presence of unconventional QC that can be easily tuned with a small magnetic field. We argue that the mechanism of QC is a strong hybridization anisotropy.

In this work we use a complete set of thermodynamic, magnetotransport and microscopic experimental techniques to probe the electronic anisotropy of $\alpha$-$YbAlB_4$, as it is driven toward two electronic instabilities with a magnetic field that is small compared to $T_0 \approx 200$ K. Most prominently, we find that the sign of the thermal expansion, which directly probes the pressure dependence of the entropy[12], changes at $B_c = 3.6$ T due to a change in the nature of the relevant fluctuation scale from magnetic correlations at lower fields, to the Kondo or valence correlations at higher fields – a signature of the proximity to a QCP. At a slightly lower field $B_c = 2.1$ T, the Shubnikov–de Haas measurements show the appearance of a new, strikingly anisotropic Fermi surface (FS), indicating a Lifshitz transition. We

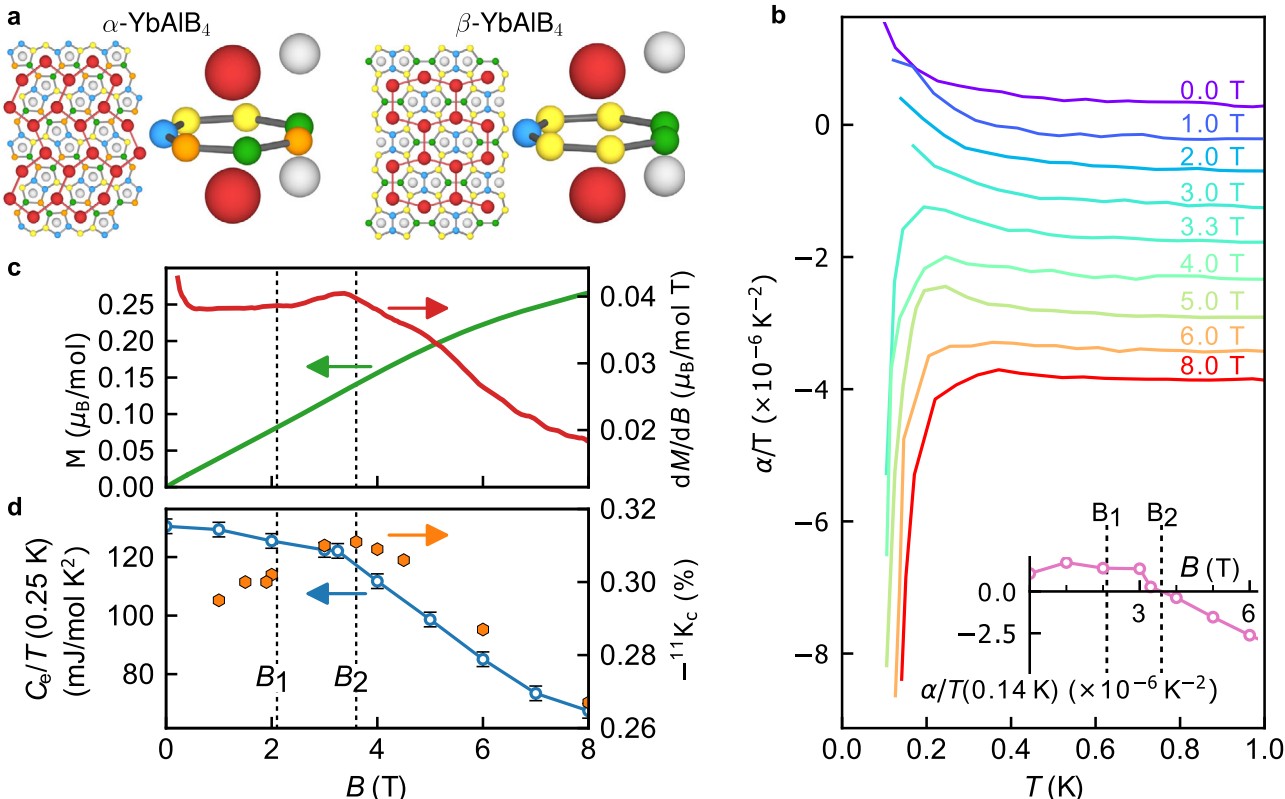

**Fig. 1 Crystal structure and magnetic field $B\|c$ axis behavior. a** Crystal structure of $YbAlB_4$ formed from straight and zigzag arrangements of hexagons of Yb atoms, and the atomic neighborhood of Yb ions — the Yb site is shown in red together with the surrounding B and Al sites. B sites are colored in blue, yellow, green and orange according to their symmetry position, while Al is gray. **b** Thermal expansion $\Delta L_c/L_c$ and thermal expansion coefficient $\alpha_c = d(\Delta L_c/L_c)/dT$ along the $c$ axis for several magnetic fields. The inset shows the field dependence of $\alpha_c$ taken at 140 mK. **c** Magnetization and susceptibility $\chi = dM/dB$ vs $B\|c$ at $T = 0.08$ K. **d** Electronic specific heat coefficient $\gamma_e$ measured at 250 mK and Knight shift of the $^{11}$B nucleus vs $B\|c$ measured at 142 mK. Vertical error bars in figure are least-square fit errors (1 s.d.).

refer to these fields as $B_1 = 2.1$ T and $B_2 = 3.6$ T in what follows. At both of these fields nuclear magnetic resonance (NMR) measurements find a diverging spin lattice relaxation rate $1/T_1$ of $^{11}B$ nuclei down to 50 mK. The magnetostriction, i.e. the rate of change of the lattice constant with magnetic field, is highly anisotropic: for the $c$ axis the magnetostriction is maximum at $B_2$, while for the $a$ axis it is maximum at $B_1$. The resistivity shows non-FL (NFL) behavior at $B_1$ and $B_2$ but only for current applied parallel to the $c$ axis. The striking two dimensionality (2D) of the FS that appears at $B_1$ is consistent with a Lifshitz transition, which was originally proposed for $\beta$-YbAlB$_4$[13], while the change in energy scales at $B_2$ indicates the proximity to a QCP. The discovery of a Lifshitz transition in an intermediate valence system at such a low field alone is remarkable, but its conjoint appearance with a QC indicates the emergence of a new physical mechanism. By combining complementary experimental techniques we propose that hybridization anisotropy, i.e. the momentum-dependent nature of the Kondo hybridization, provides the means for these phenomena to manifest at the small energy scales associated with fields $B_1$ and $B_2$ in contrast to the large intermediate valence energy scale. Our conclusion is supported by recent experimental verifications of anisotropic hybridization in YbAlB$_4$ by ARPES[14] and linear dichroism in core-level HAXPES[15], but also by the unique properties of the thermoelectric power factor[16].

## Results

**Ground state properties from macroscopic probes.** We first describe the magnetic field dependence of the thermal expansion $\Delta L_c / L_c$, where $L_c$ is the sample length along the $c$ axis. At $B = 0$ T and low temperatures ($T < 0.2$ K) the linear thermal expansion coefficient $\alpha_i$ is positive for both the $i = a$ (not shown) and $c$ axes (Fig. 1b), and the volume coefficient $\alpha_{Vol} > 0$ already for $T < T^*$ ≈ 8 K. This is surprising since usually in Yb-based Kondo-lattice (KL) or intermediate valence systems $\alpha_{Vol} < 0$, just like the magnetostriction $\lambda_i = d(\Delta L_i / L_i)/dB$ (see Supplementary Note 1). However, $\alpha_{Vol}$ measures the pressure dependence of entropy, which in $\alpha$-YbAlB$_4$ implies that the dominant contribution does not arise from KL or intermediate valence type fluctuations, but from an energy scale that increases with pressure. In magnetic Yb-based systems this is usually (anti-)ferromagnetic (A)FM order mediated by the RKKY interaction[17,18]. In $\alpha$-YbAlB$_4$ the high Wilson ratio[9] ($\chi_0 / \gamma_0 \approx 7$) and $\alpha_{Vol}(B=0) > 0$ indicate the presence of FM correlations in the ground state. It has been found that pressure induces an AFM state in $\beta$-YbAlB$_4$ through a first-order phase transition at 2.5 GPa[19,20], and an AFM state emerges in $\alpha$-YbAlB$_4$ by Fe doping[21,22] at 1.5%.

Under magnetic field the magnetic correlations are suppressed and the sign of $\alpha_c$ changes between 3.3 T and 4 T, indicating a change of the relevant energy scale as expected at a crossover or a phase transition between an ordered and a disordered phase. At a QCP it is expected that $\alpha_{Vol}/T$ besides changing sign[12] also becomes divergent. However, in $\alpha$-YbAlB$_4$ this change is rather smooth and asymmetric, similar to the QCP found in YbAgGe[23], and the system remains paramagnetic (PM) on both sides of $B_2$. This behavior is typical of metamagnetic materials, like Sr$_3$Ru$_2$O$_7$[24] or CeRu$_2$Si$_2$[25] where the entropy is dominated by magnetic fluctuations (see Supplementary Note 1). The relatively smooth change in $\alpha_{Vol}/T$ suggests that $\alpha$-YbAlB$_4$ is located either close to a quantum critical endpoint, or in the proximity of a field-induced QCP. Within both scenarios a clear anomaly in magnetic susceptibility $\chi$ and specific heat coefficient $\gamma$ is expected. The magnetization $M$ and $\chi = dM/dB$ measured with $B\|c$ are shown in Fig. 1c, with $dM/dB$ displaying a clear enhancement at $B_2$ before decreasing rapidly at higher magnetic

field. However, compared with true metamagnetic materials the enhancement is weak and is not symmetric around $B_2$. The same features are seen in the specific heat coefficient $\gamma$ and in the NMR Knight shift $^{11}K$ measured at the $^{11}B$ nucleus, as shown in Fig. 1d. Since all three quantities are governed by the Fermi surface properties: $\gamma \propto \chi \propto {}^{11}K \propto N(\epsilon_F)$, with $N(\epsilon_F)$ the density of states at the Fermi level $\epsilon_F$, it is clear that the anomaly at $B_2$ involves a continuous reduction of $N(\epsilon_F)$ as a consequence of the suppression of the correlations by the magnetic field. This is different from a possible suppression of the HF state through a metamagnetic transition. In fact, when compared to HF metamagnetic compounds[26] the value of $B_2$ is not large enough. In $\alpha$-YbAlB$_4$ the relevant magnetic field scale, estimated from $\gamma = 130$ mJ/molK$^2$, should exceed 20 T, and hence metamagnetism cannot account for the observed phenomena. Therefore, the QC signatures in $\alpha$-YbAlB$_4$ are unconventional. One of the ways we could confirm a QCP would be to find a diverging Güneisen parameter[23] ($\Gamma \propto \alpha_{Vol}/C_e$). Unfortunately, a large nuclear contribution[22] in specific heat upon subtraction introduces a great uncertainty in the Güneisen parameter. Hence, we turn to other techniques to check the nature of the transition at $B_2$.

**Intrinsic excitations tuned by magnetic field.** A rigorous test of the ground state properties is the character of underlying excitations. To probe them we have measured the NMR relaxation rate $1/T_1$ (see Methods) of the $^{11}B$ nuclei (plotted in Fig. 2a, b) as a function of temperature at various magnetic fields for $B\|c$. The data are shown as $(T_1 T)^{-1}$ so that they can be easily contrasted to $(T_1 T)^{-1} =$ const. typical for FL state of simple metals at low temperature. Here we observe the FL state up to 1.5 T and below $T^*$, where $(T_1 T)^{-1}$ is ≈0.7 K$^{-1}$s$^{-1}$. As the magnetic field approaches $B_2$, we surprisingly find a pronounced divergence $(T_1 T)^{-1} \propto T^{-\delta}$ at low temperatures with exponent $\delta = 0.36$. This power-law behavior persists down to 50 mK, two orders of magnitude lower than $T^*$, and reveals the presence of quantum critical fluctuations that destabilize the FL and lead to a NFL ground state. Across $B_2$ the NMR spectrum remains unchanged (Supplementary Figs. 1, 2a, b) which indicates no magnetic or charge order appears at $B_2$, as the spectra would otherwise split due to symmetry breaking.

We have also analyzed the Knight shift behavior $K = \langle b_z \rangle / B$, where $\langle b_z \rangle$ is the hyperfine magnetic field at the nucleus site. While $(T_1 T)^{-1}$ is sensitive to spin excitations at $\mathbf{q} \geq 0$, the Knight shift is related to the static spin susceptibility $\chi(\omega = 0, \mathbf{q} \approx 0)$, and for a FL it is expected to be constant against temperature and magnetic field. As seen in Fig. 1d (and Supplementary Figs. 3 and 4), for $B < B_2$ the $c$-axis Knight shifts of $^{11}B$ ($^{11}K_c$) and $^{27}Al$ ($^{27}K_c$) nuclei are constant, and at $B_2$ show a weak feature (similar to $\chi$) confirming the absence of magnetic $\mathbf{q} = 0$ mode at $B_1$ and $B_2$. Additionally, the NMR linewidth (Supplementary Fig. 2) is also consistent with the absence of magnetism. Across 4 T, both $^{11}K_c$ and $^{27}K_c$ show only a slow drop arising from $N(\epsilon_F)$. This lack of prominent static changes is also supported by measurements of the quadrupolar coupling of $^{27}Al$ and $^{11}B$ (Supplementary Fig. 5) that show only a small change.

So far, we have focused on the QC at $B_2 \approx 3.6$ T visible in thermodynamic and NMR measurements. In addition, the NMR measurements show the existence of another unusual behavior that suddenly appears at a lower field $B_1 \approx 2.1$ T. Indeed, the magnetic field dependence of the NMR relaxation rate $(T_1 T)^{-1}$ at 142 mK shows two pronounced peaks (Fig. 2a) at $B_1$ and $B_2$. The temperature dependence reveals another NFL power-law divergence at $B_1$, $(T_1 T)^{-1} \propto T^{-\delta}$, with $\delta = 0.25$ similar to the behavior at $B_2$. Between $B_1$ and $B_2$ the system behaves as a standard metal, down to lowest temperatures, which indicates $B_{1,2}$ are separate

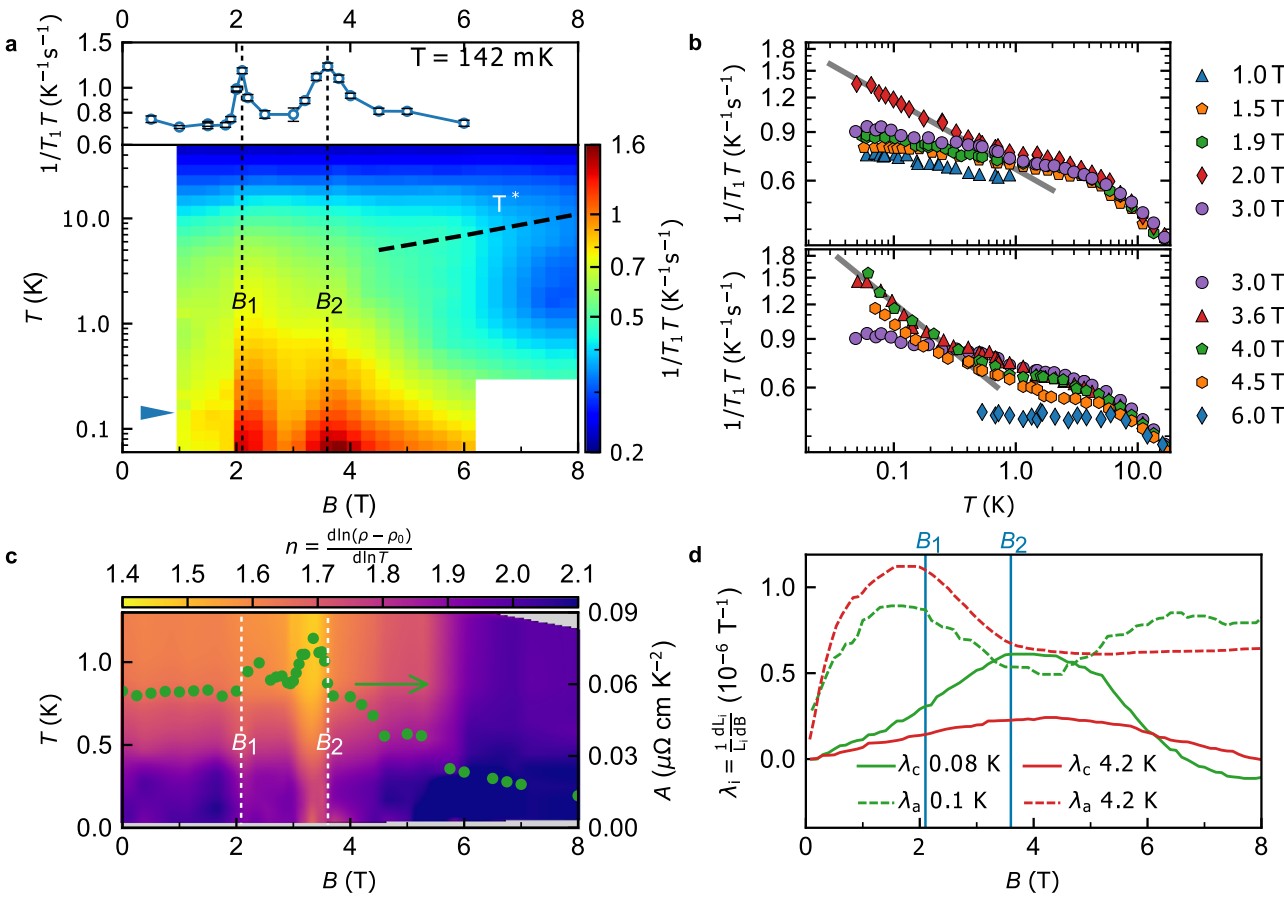

**Fig. 2 Detection of quantum criticality by NMR, resistivity and magnetostriction. a** (Lower panel) Density plot forming a $B − T$ phase diagram of $(T_1T)^{-1}$ on $^{11}$B sites showing the divergent behavior of spin fluctuations at the two critical fields $B_1 = 2.1$ T and $B_2 = 3.6$ T. (Upper panel) Cross section of $(T_1T)^{-1}$ on $^{11}$B sites at $T = 142$ mK as indicated by the arrow in the lower panel. Vertical error bars are least-square fit errors (1 s.d.). **b** Temperature dependence of $(T_1T)^{-1}$ at fields close to $B_1$ (upper panel) and $B_2$ (lower panel). Error bars are equal or smaller than symbol size, and they correspond to least-square fit errors (1 s.d.). The lines show a power law dependence of $\propto T^{-0.25}$ (upper panel) and $\propto T^{-0.36}$ (lower panel). The slight mismatch between the $(T_1T)^{-1}$ values in (**a**) comes from a ~ 1° deviation in magnetic field orientation with respect to the $c$ axis. **c** Density plot of the power law exponent of resistivity along the $c$ axis and the $A$ coefficient extracted at $T = 0.1$ K vs $B||c$. **d** Magnetostriction coefficient $\lambda$ vs $B$ along both $a$ and $c$- axes at $T = 0.1$ K and $T = 4.2$ K.

phenomena. Their qualitatively different nature is evidenced by the critical dynamics at $B_1$ that, unlike the one at $B_2$, shows no distinct signature in susceptibility or specific heat despite clear evidence of divergent spin fluctuations from the NMR.

Further indications about the nature of the transition at $B_1$ are given by the magnetostriction coefficient $\lambda$, which is sensitive to structural, magnetic and electronic structure transitions, measured along the $a$ axis ($\lambda_a$) and $c$ axis ($\lambda_c$) with magnetic field $B||c$ (Fig. 2d). It can be seen that it shows a strong anisotropy: while $\lambda_c(B)$ shows a maximum at $B_2$ as expected from $\alpha_c$, $\lambda_a(B)$ shows a maximum at the smaller field $B_1$, which implies that it is related to a 2D effect within the $ab$ plane.

Anisotropy in the critical behavior of $\alpha$-YbAlB$_4$ is also evidenced by resistivity measurements along the $a$ and $c$ axes, $\rho_a$ and $\rho_c$, respectively. Low-temperature NFL behavior due to QC is quantified by observing the exponent $n_i$ and the coefficient $A_i \propto m_i^2$ as $\rho_i = \rho_{0,i} + A_i T^{n_i}$, $i = a, c$ (see Methods). At $T = 0.1$ K, $A_c$ shows an enhancement (Fig. 2c) by $\approx 50\%$ at $B_1$ and $B_2$ before decreasing at higher fields, and the exponent $n_c$ shows deviations (Fig. 2c) from the FL value of 2 in the vicinity of $B_1$ and $B_2$, where $n_c(B_2) = 1.65$ at the lowest measured temperature $T = 0.04$ K. Although this value is close to that expected at a FM or AFM QCP where $n = 5/3$ and $3/2$, respectively[27], since no magnetic order is observed, the QC of the pristine sample arises with the suppression of magnetic correlations. By contrast, $\beta$-YbAlB$_4$

shows $n_{ab} = 3/2$ in $\rho_{ab}$ at its critical point consistent with the AFM QCP (even though fluctuations dominate[28] $\chi(q \approx 0)$). Note that the NFL behavior in $\alpha$-YbAlB$_4$ occurs only for $c$-axis resistivity $\rho_c$, whereas $n_a$ remains $\approx 2$ for $\rho_a$ (Supplementary Fig. 8). This shows that QC originates from specific regions of momentum space, in accordance with the anisotropic magnetostriction and the NMR data. Intrinsic electronic anisotropy is therefore a pivotal factor for understanding this system.

**Properties of the Fermi surface.** We now consider the changes in the electronic structure in the vicinity of $B_1$ and $B_2$ by measuring quantum oscillations (QO). Figure 3a shows $\rho_c$ as a function of the magnetic field, from which we separate the slow-varying ($\rho_{MR}$) and the oscillatory ($\rho_{osc}$) component (see Methods). A clear kink in $\rho_{osc} \times B^{1/2}$ (Fig. 3b) is observed at $1/B_1 \approx 0.5$ below which oscillations appear. The logarithmic scale is used to show that the amplitude decays linearly as expected from the Dingle relation. The amplitude is also shown against temperature in Fig. 3c, together with a fit to the Lifshitz-Kosevich (LK) relation, that describes the decay of QOs with $T$ and gives $m^* = 0.55 \pm 0.03$. The excellent agreement to the LK relation confirms these are indeed QOs with frequency $F = 10.2$ T, while their sudden appearance at $B_1$ suggests they emerge as the result of a Lifshitz transition at $B_1$. This remarkable result makes $\alpha$-YbAlB$_4$ the first

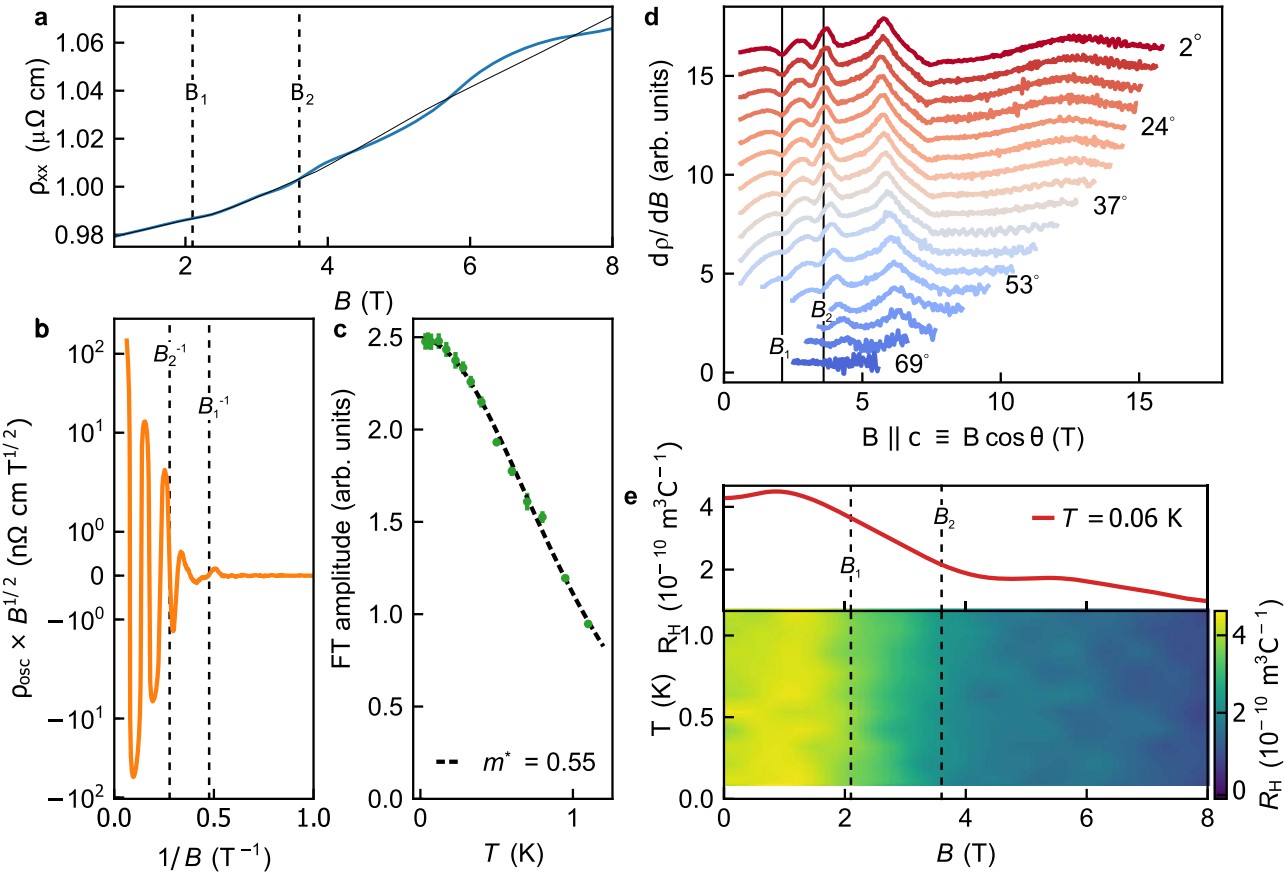

**Fig. 3 Quantum oscillations and Hall effect in $\alpha$-YbAlB$_4$. a** $\rho$, $I\|c$ vs $B\|c$ at $T = 0.03$ K, the overlaid black line is the non-oscillatory component, $\rho_{MR}$ that is subtracted. **b** The oscillatory component of resistivity; $\rho_{osc} \times B^{1/2}$ plotted on a double logarithmic scale (values $< 10^{-3}$ are plotted on a linear scale). **c** Temperature dependence of the Fourier transform of $\rho_{osc}$. **d** $d\rho/dB$ against $B\|c$ at various angles of $B$ in the [001] → [110] plane where $\theta = 0 \equiv B\|$[001]. **e** (Upper panel) Hall coefficient $B\|c$ at $T = 0.06$ K. (Lower panel) surface plot of Hall coefficient against field and temperature.

intermediate valence compound with a Lifshitz transition induced by such a low magnetic field.

By rotating the sample relative to $B$ we probe the extremal cross-section of this FS perpendicular to $B$. Figure 3d shows $d\rho/dB$ against the $c$-axis component of $B$, i.e. $B\cos\theta$ where the $B$ is rotated in the plane of [001] and [110]. The $\rho_{osc}$ as a function of $B\cos\theta$ remains unchanged for different field directions – indicating the FS is cylindrical and arises from 2D carriers, with a very small volume of this cylinder of $4 \cdot 10^{-4}$ carriers per Brillouin zone. The 2D FS that emerges at $B_1$ confirms the role of anisotropy in the electronic environment of the system. The QOs of different spin components of a 2D FS are normally split by the Zeeman interaction and interfere to produce spin zeros where the QO amplitude vanishes and the phase shifts by $\pi$. Using a $g$ factor obtained by electron spin resonance[29] and the determined value of $m^*$, we would expect a strong angular dependence of the amplitude and a spin zero at $\theta = 60°$ (see Supplementary Note 3). However, no spin zeros are found, suggesting that this pocket is spin polarized. Taking $m_J = 5/2$, as deduced by theory[30] and shown by experiment[15], the magnetic moment of this pocket is $0.002\mu_B$, which is consistent with extremely small polarization increase observed in $M$ in Fig. 1c. Therefore, considering both the primary evidence from QOs together with the consistency with macroscopic measurements, the results overwhelmingly indicate a Zeeman driven Lifshitz transition occurs at $B_1$.

The 2D nature of the FS pocket, the divergence in $(T_1T)^{-1}$ at the $^{11}$B site and the pronounced maximum in $\lambda_a$ indicate that due to anisotropy the electronic properties within boron layers are most affected at $B_1$. However, the small size of the FS pocket

relative to the total number of carriers makes it difficult to account for the strong quantum critical fluctuations at $B_2$ and suggests that larger sheets of the FS are affected. While large QO frequencies are observed at higher magnetic fields (see Supplementary Note 3), these were difficult to observe at low magnetic fields of $B_1$ and $B_2$, and we therefore turn to the Hall coefficient ($R_H$) shown in Fig. 3e. $R_H$ decreases smoothly above $B_1$, which would within a single-band model indicate a change in the carrier density by a factor of 2 with respect to the $R_H(B = B_2)$ value. However, as this is not supported by the small changes in $\rho$, it could only be interpreted as changes in the FS velocity on larger sheets of the FS, and not as a second Lifshitz transition at $B_2$. The $(T_1T)^{-1}$ data, the anisotropic resistivity, sign change of thermal expansion and the gradual change in $R_H(H, T)$ therefore confirm that $\alpha$-YbAlB$_4$ is close to a QCP at the field $B_2$.

## Discussion

Intermediate valence compounds have been mainly outside the main focus of research on QC since, due to large characteristic energy scales of the valence fluctuations (e.g., $T_0 > 400$ K in YbAl$_3$), application of large magnetic fields or pressures is required to notably modify their properties or induce new phases[31]. In contrast, for both YbAlB$_4$ polymorphs magnetic moments survive well below $T_0 \approx 200$ K, and become strongly correlated below $T^* \approx 8$ K, showing QC at low magnetic fields. Although the quantum critical behavior in the two polymorphs has different phenomenology, the pervasiveness of highly anisotropic Kondo hybridization in both compounds appears to be

crucial for establishing it. In particular, the presence of a quantum critical behavior in different conditions for both, $\beta$-YbAlB$_4$ (under ambient conditions, with doping and under increased pressure) and in $\alpha$-YbAlB$_4$ (without doping but under magnetic field, and with doping), leads to the conclusion that in YbAlB$_4$ system, in the presence of strong electronic anisotropy, a robust physical mechanism emerges; one insensitive to the change of local symmetry and chemistry. The only energy scale that matches this description is the anisotropic Kondo hybridization, established by the crystal structure where Yb chains interpenetrate[28] the sheets of B. The conventional picture of intermediate valence materials contains the hybridization energy scale $\Gamma = \pi N(\epsilon_F)|V|^2$, where $V$ is Anderson's coupling strength between conduction band and localized $f$-electrons. In the strong coupling limit $\Gamma$ approaches $T_K$. The NFL behavior and Kondo hybridization can be connected[30] by appealing to the tensorial, momentum-dependent nature of the hybridization stemming from the dominant $m_J = \pm 5/2$ nature of Yb ground state doublet in the form of $H_V = \hat{V}_{\sigma\alpha}(\mathbf{k})c_{\mathbf{k}\sigma}^{\dagger}f_{\mathbf{k}\alpha} + $ h.c. , expressed in terms of the creation/annihilation operators of conduction ($c^{\dagger}$) and localized ($f$) electrons. Subsequent theoretical work on $\beta$-YbAlB$_4$ showed[13] that the tensor $\hat{V}_{\sigma\alpha}(\mathbf{k})$ is indeed highly anisotropic and vanishes upon approaching the $\Gamma - Z$ line in the Brillouin zone as $|\hat{V}_{\mathbf{k}}| \propto \sin(k_z c)k_{\perp}^2$, which is also confirmed by recent ARPES[14] and linear dichroism in core-level HAXPES[15] measurements. In the lower symmetry polymorph $\alpha$-YbAlB$_4$, this dispersion is not expected to persist, however, the observation of a Lifshitz transition of the strikingly 2D FS pocket and the deviation from FL behavior suggest that $B_1$ is remnant of the QCP in $\beta$-YbAlB$_4$, but modified, as $\rho_{ab}$ in the two compounds show strikingly different behavior. In the case of $\alpha$-YbAlB$_4$, $B_1$ is detuned from zero field by a non-zero value of the renormalized chemical potential $\epsilon_F^*$: $\xi(\mathbf{k}) = \epsilon_F^* - g\mu_B B + N(\epsilon_F) \text{Tr}(|\hat{V}(\mathbf{k})|^2)$, whereas $\epsilon_F^*$ is believed to be zero[13] in $\beta$-YbAlB$_4$. This is consistent with the experimental data in $\alpha$-YbAlB$_4$ if $\epsilon_F^*$ is $g\mu_B B_1 \approx 2$ meV, such that the Lifshitz transition appears at $B_1 = 2.1$ T, when the chemical potential reaches the bottom of the majority-spin band.

Unlike the transition at $B_1$, the NFL behavior at $B_2$ is a much more 3D phenomenon that drastically changes the global properties of the system. The FM fluctuations present in the ground state at zero field (shown by $\alpha_{\text{Vol}} > 0$ and a high Wilson ratio) are suppressed as the magnetic field is increased to $B_2$. Near $B_2$, although FM ($\mathbf{q} \sim 0$) fluctuations are suppressed, finite-$\mathbf{q}$ fluctuations get enhanced, causing power-law divergence of $(T_1 T)^{-1}$, NFL physics in resistivity and a sign change of thermal expansion coefficient, which imply the proximity to a QCP unrelated to the magnetic one in Fe-doped system. At the same time, magnetostriction, QO and Hall effect show that the transition is followed by a FS change. The QCP also differs from the one in $\beta$-YbAlB$_4$, where the well understood FS of $\beta$-YbAlB$_4$[13,14,32,33] has no small pocket like the one observed to appear at $B_1$ in $\alpha$-YbAlB$_4$, and signatures of spin zeros were observed on larger FS sheets[34] which are absent in the $\alpha$-YbAlB$_4$. Although in intermediate valence compounds magnetic field can induce a valence QCP[35], our measured data exclude this for the case of $\alpha$-YbAlB$_4$ (see Supplementary Note 2). Recently[36], Mössbauer spectroscopy found a change in the quadrupolar moment of Yb between 1 and 5 T. This would imply that the QC at $B_2$ originates from a multipole-type QCP in $\alpha$-YbAlB$_4$. However, due to the low symmetry of the Yb site, its understanding will require further measurements which are beyond the scope of the current work.

It is remarkable that (coherent) critical fluctuations survive the large energy scale $T_0$ – particularly since there has been only one reported quantum critical intermediate valence compound so far: CeRhSn[37]. Although it is also anisotropic, there the 2D frustrations

drive the QC, while this is not likely in the case of $\alpha$-YbAlB$_4$. The unconventional nature of QC in $\alpha$-YbAlB$_4$, where a divergence is found in the dynamical quantity $T_1$, but without observing symmetry breaking, shows that intermediate valence compounds can also offer novel types of critical phases. The tensorial nature of the Kondo hybridization and its vanishing at certain high-symmetry points in the Brillouin zone is also a central thesis of the theory of topological Kondo insulators[38], and has shown promising results in a recent theoretical treatment of the quasi-1D Kondo lattice[39]. However, a more detailed analysis is required to encompass the richness of the phase diagram of the YbAlB$_4$ family.

From all the data shown above we can conclude that hybridization nodes offer a new route of overcoming the high energy scale of intermediate valence compounds resulting in quantum criticality and new phases of matter.

## Methods

**Sample synthesis**. Single crystals of $\alpha$-YbAlB$_4$ were grown from Al flux. The stoichiometric ratio of Yb:4B was heated in excess Al in an alumina crucible under an Ar atmosphere as described elsewhere[40]. Chemical compositions of single crystals were determined by a inductively coupled plasma - atomic emission spectrometry (ICP-AES, HORIBA JY138KH ULTRACE) at ISSP, and the analysis of both polymorphs are in good agreement with the ideal compositions of YbAlB$_4$ within the error bars. We analyzed diffraction patterns to determine the crystal structure and lattice constant using the Rietveld analysis program PDXL (Rigaku) and found no impurity phase.

**Experimental details**. The thermal expansion and magnetostriction were measured with a high-resolution capacitive CuBe dilatometer in a dilution refrigerator[41]. Specific heat was measured using relaxation calorimetry[42]. Further detail and subtraction of nuclear contributions will be described elsewhere. Magnetization was measured using capacitance Faraday method[43].

Resistivity and Shubnikov-de Haas measurements were performed using conventional lock-in amplifier techniques. For measurements with $I\|c$ pristine crystals were used, while for $I\perp c$ and for Hall effect measurements larger crystals were polished to form thin plates perpendicular to the $c$ axis. Low temperature deviations from Fermi liquid (FL) behavior due to quantum critical fluctuations are quantified by expressing $\rho_i = \rho_{0,i} + A_i T^{n_i}$. Taking $n = 2$ we extracted $A_i(T) \propto m_i^2$. Similarly, we extracted the temperature exponent by assuming $A$ is constant in $T$ as $n_c = d\ln\delta\rho_c / d\ln T$.

For quantum oscillations the small number of oscillation periods made the extraction of the oscillatory component of the resistance challenging; we applied two methods that gave consistent results. We assume that $\rho = \rho_{\text{MR}} + \rho_{\text{osc}}$ where $\rho_{\text{MR}}$ is assumed to be slowly varying. In the first method we apply a low pass filter in $1/B$ that subtracts a locally quadratic polynomial, this is described in detail elsewhere[44]. In the second we first subtract a linear component from the entire field range and then take a derivative. A comparison between the Fourier spectrum of these methods is shown in the supporting information (Supplementary Note 3 and Supplementary Fig. S-11).

The NMR measurements of $^{11}$B were performed using a pulsed spectrometer. The spectra were collected by Hahn echo sequence $\pi/2 - \tau - \pi$ with typical value of $\tau = 100$ µs, and a $\pi/2$ pulse of 6 µs. The $T_1$ measurements were performed using a saturation-recovery technique on a satellite NMR line determined in previous work[45]. Data were fitted to a magnetic relaxation function of the form: $M(t) = M_0(1 - (1/10)e^{-(t/T_1)^{\beta}} - (5/10)e^{-(3t/T_1)^{\beta}} - (4/10)e^{-(6t/T_1)^{\beta}}$. The sample was oriented in situ by observing the quadrupolar splitting of $^{27}$Al using a two-axis goniometer for measurement temperatures 1.5–300 K, and with a single-axis goniometer for measurements at lower temperatures. In all cases the magnetic field orientation was within 2° of the crystal $c$ axis.

## Data availability

The datasets generated and/or analyzed during the current study are available from the corresponding author on reasonable request.

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

## Acknowledgements

We acknowledge the help of Naoki Horie. M.S.G. acknowledges the support of Croatian Science Foundation (HRZZ) under the project IP-2018-01-2970, the Unity Through Knowledge Fund (UKF Grant No. 20/15) and the support of project CeNIKS co-financed by the Croatian Government and the European Union through the European Regional Development Fund - Competitiveness and Cohesion Operational Programme (Grant No. KK.01.1.1.02.0013). The work at Rice University (A.H.N.) was supported by the U.S. National Science Foundation CAREER grant no. DMR-1350237 and the Welch Foundation grant C-1818. R.K. and M.B. are supported by the German Science Foundation through Projects No. BR 4110/1-1 and No. KU 3287/1-1, respectively. M. L. S. acknowledges the support of the EPSRC and the Royal Society. The work at the Institute for Quantum Matter, an Energy Frontier Research Center was funded by DOE, Office of Science, Basic Energy Sciences under Award # DE-SC0019331. This work was partially supported by JST-CREST (JPMJCR18T3), Japan Science and Technology Agency, and Grants-in-Aid for Scientific Research (19H00650). S.N. acknowledges support from the CIFAR as a Fellow of the CIFAR Quantum Materials Research Program. The use of the facilities of the Materials Design and Characterization Laboratory at the Institute for Solid State Physics and the Cryogenic Research Center, The University of Tokyo, is gratefully acknowledged.

## Author contributions

E.C.T.O., M.S.G. and Y.M. contributed equally to this work. E.C.T.O. performed the resistivity, Hall effect and quantum oscillations measurements, M.S.G., M.Y. and M.T. performed NMR experiments, Y.M. performed specific heat measurements. Y.M., K.K., Y.K., Y.S. and T.S. performed magnetization measurements and K.K. synthesized the samples. R.K. and Y.M. performed the thermal expansion and magnetostriction measurements. M.L.S., M.B. and A.H.N. contributed to data interpretation. M.T. and S.N. conceived the project, planed the research and contributed to data interpretation. E.C.T.O., M.S.G. and Y.M. wrote the paper. All authors took part in discussing results and editing the manuscript.

## Competing interests

The authors declare no competing interests.
