## [Peer Review File · Nature Communications]

Reviewers' comments:

Reviewer #1 (Remarks to the Author):

The manuscript submitted by Grbić et al. consists of a detailed study of an interesting compound α -YbAlB₄ by several measurements (resistivity, thermal expansion, NMR, magnetization and specific heat).

Quantum criticality has been a topic of discussion and exploration for a long time. In this matter, Yb-based heavy fermion systems are of special interests. Special is this two brother compounds (α -YbAlB₄ and β -YbAlB₄). β -YbAlB₄ has been explored a lot because of its fascinating characteristics. Whereas, this present manuscript deals with the other sibling α -YbAlB₄. The authors have found quantum critical behavior along with a Lifshitz transition under low applied magnetic field and attributed the quantum critical behaviors to the anisotropy of hybridization.

The referee listed out some comments / suggestions / questions which are the followings -

1. The referee would suggest to indicate the critical fields (B1 and B2) in all the field-dependent plots which will be helpful for the readers.
2. Why the authors did not show Grüneisen ratio which is a better parameter to see quantum criticality. Such a plot will be necessary to support the claim of the authors.
3. The referee would suggest to plot $\alpha c/T$ as a function of temperature in the place of figure 1(b).
4. Is there are any possibility of having Lifshitz transition even at B2 along with B1? The referee is curious because the evidence of Lifshitz transition was claimed to be seen only from the kink in Fig 3(b), so if there is also a Lifshitz transition at B2, then that would be merged with the oscillations. As a result both at B1 and B2 a change, though small, has been seen in the 11vQ.
5. Page 5, 3rd paragraph: Only from the magnitude of $1/T1T$ it is not reasonable to say about the closeness to magnetism, especially comparing the magnitude with other compounds where the hyperfine coupling constants are different.
6. The referee would suggest include the power-law lines in figure 2(b) of $1/T1T$.
7. The temperature dependence of spectra, K (%) and line width are important to show down to low temperatures along with the spectra as a function of the field at the lowest temperature to nullify the possibility of having short-range or long-range ordering at low temperature and across the critical fields. Furthermore, the $1/T1T$ vs K (%) plot will be helpful to uncover the nature of correlations (FM/AFM). In this respect, the sentence "As seen....B2" in 6th Page 2nd paragraph seems also not a proof of the absence of $q=0$ mode as most of the features are rather weak.

8. In general, as the authors know that the spin-lattice relaxation process along c direction is driven by the spin-fluctuations perpendicular to c -direction, so to have an understanding about the critical fluctuations along different directions, the referee would suggest to estimate the spin-fluctuations only along c -axis for which relaxation measurements perpendicular to c is essential. This is of special importance as the system is highly anisotropic as far as the criticality is concerned.

9. Page 5, 2nd paragraph: The exponent of $1/T_1T$ is indeed very small but also close to e.g. YbRh_2Si_2 (0.5) near QCP and the low value of exponent may occur due to other reasons as well, not reasonable to claim that the criticality is nonmagnetic only from that exponent. To claim a close proximity to magnetic order, the unchanged spectra across B_1 and B_2 is important, if so. Though low-temperature spectra are not shown and should be shown. Furthermore, it also mimics the behavior with a slow increase of $1/T_1T$ in case of CeFePO at a metamagnetic transition with a possibility of fermi surface instability (PRL 107, 277002 (2011)). Also see the case for $\text{Sr}_3\text{Ru}_2\text{O}_7$ (PRL 95, 127001 (2005)). Can the authors put light on the possibility of such a scenario in this system? It is more likely that the B_1 and B_2 are metamagnetic transitions similar like the mentioned systems. Again, the referee believes that it is really important to estimate spin-fluctuations only along c -axis to say something concretely about the power law and the divergences of $1/T_1T$ at B_1 and B_2 .

10. From $1/T_1T$ data, it is at all not the case that in the field range between B_1 and B_2 , $1/T_1T$ behaves as a standard metal because $1/T_1T$ at 3 T shows a clear indication of critical fluctuations. This might suggest that B_1 and B_2 might not also be two separated phenomena.

11. With a more closer look at Figure 2(a): why the magnitude of $1/T_1T$ at 142 mK have been plotted, it should be the lowest temperature measured. In that case, the value at 4 T is higher than that of at 3.6 T. How about the stretch exponential in fitting the recovery curves? Can the authors say something about the error in determining T_1 ? The referee would suggest to include some recovery curves at low temperatures in the supplemental materials.

12. Can the authors say somethings about the fact that frustration is not at all affecting these critical behaviors? The referee finds it rather important, also to explain the anisotropic nature of criticality, especially in case of thermal expansion coefficient, as have been seen in CeRhSn (Sci. Adv. 1, e1500001 (2015)). The crystal structure $\alpha\text{-YbAlB}_4$ has, if one considers the second nearest neighbor exchange (which seems to be reasonable to consider), then frustration can appear.

13. The referee also would like to mention that the effect of anisotropy in quantum criticality has been seen also in other systems e.g. CeRhSn and also in case of Fe doped CeRuPO (J. Phys. Soc. Jpn., 82, 033704 (2013)).

Final remarks: The referee is convinced with the fact that the systems are indeed very interesting and provides a plethora of features to cultivate. The manuscript may be accepted at Nature communications only after providing reasonable answers / clarifications of the queries raised by the referee.

Reviewer #2 (Remarks to the Author):

The manuscript “Anisotropy-driven quantum criticality in an intermediate valence system” by Grbic et al presents thermodynamic, magnetotransport and microscopic experiments on α -YbAlB₄ in an applied magnetic field (B). The authors observe two features at $B_{c1}=2.1$ T and $B_{c2} = 3.6$ T and associate them with a Lifshitz transition and a quantum critical point, respectively. The authors then attribute the quantum critical behavior to the anisotropic hybridization between f- and conduction electrons. Efforts into determining the origin of quantum critical behavior in strongly correlated systems are important, and the breadth of low-temperature techniques reported in this manuscript is remarkable. There are, however, important questions that need to be answered to demonstrate the authors’ claims and to meet Nature Communications’ criteria:

- 1) A central point of the manuscript is that the origin of quantum criticality is the anisotropic hybridization between f- and conduction electrons. I do agree that anisotropic hybridization is a key parameter in f-based systems because it is a property of the ground state wavefunction which inevitably determines Kondo and RKKY interactions. Anisotropic hybridization per se, however, cannot generate quantum criticality, but it rather allows for the fluctuations that drive criticality. The fundamental parameter to characterize quantum criticality is a characteristic length scale which diverges at the quantum critical point. What is this length scale in α -YbAlB₄?
- 2) Anisotropic hybridization also is relevant for the formation of a heavy-fermion state and, if the state at about B_{c2} has Kondo correlations, then one would expect enhanced effective masses. The authors, however, obtain a light effective mass ($m^* = 0.55 m_e$). This is a contradiction that the authors need to address.
- 3) Related to the second point is the theoretical remark that the f electron appears to become detached from the Fermi energy above 3 T in α -YbAlB₄ (Ref. 12 of the manuscript). In this case, quantum critical fluctuations would not be expected
- 4) The authors state that minor changes in valence are observed to 40 T, but they fail to mention these x-ray absorption measurements were performed at 2K, which is above the region in which the effects reported here are pronounced. This is misleading and should be clarified in the manuscript.
- 5) No strong evidence for a Lifshitz transition is presented. The authors do seem to appreciate this issue initially stating that their results indicate a Lifshitz transition, but then a much stronger statement is made: “The discovery of a Lifshitz transition in an intermediate valence system at such low field alone is remarkable (...)”. This is also misleading. The authors should soften their statement as a change in the density states in field does not guarantee a Lifshitz transition.
- 6) Another important aspect of anisotropic hybridization is the actual shape of the ground state doublet wavefunction. The authors, however, do not discuss in detail the groundstate wavefunction

of α -YbAlB₄ and its comparison to β -YbAlB₄. These two systems are locally isostructural but have distinct groundstates (e.g. Ref 14 of the manuscript).

7) Further, the crystal field splitting between ground state and first excited crystal field doublet could also shed light on the evolution of the groundstate wavefunction in magnetic fields because Zeeman splitting may mix different doublets. How are the wavefunctions expected to change in field?

8) The authors state that the presence of f-electron valence fluctuations to a non-magnetic state is regarded (as) an anathema to quantum criticality. I tend to disagree with such strong statement as this is not necessarily the case. For instance, CeCu₂Si₂ and PuCoGa₅ have been argued to display superconductivity around a valence quantum critical point.

Reviewer #3 (Remarks to the Author):

This manuscript reports on experimental studies of the mixed-valence material YbAlB₄. The authors have used quite an impressive range of experimental techniques to perform thermodynamic measurement and by analyzing their experimental data they argue that the alpha-polymorph of YbAlB₄ exhibits quantum critical behavior and Lifshitz transition under an application of low magnetic fields. The authors argue that α -YbAlB₄ is unique in that it provides the first example of a system in which quantum criticality co-exists with the mixed-valence. Furthermore, the authors argue that this co-existence is governed by the momentum anisotropy of the hybridization matrix element between the f- and d-orbitals of the Yb ions.

First of all, I would like to note that the several authors of the current manuscript (K. Kuga, S. Nakatsuji, A. Nevidomskyy to name a few) have published their research on this and related materials before. In fact, the first publications date back to almost 10 years ago. Despite that history, the topical has never emerged as a 'main stream' topic, unlike the one of topological Kondo insulators (which is mentioned in the discussion section of the manuscript). From this, I tend to think that despite the current efforts, the earlier and present experimental findings while remaining interesting for the experts in the field of f-orbital based ("heavy-fermion") materials, will unlikely bring this topic into the main-stream of the contemporary condensed matter physics.

While the experimental analysis is broad and appears to be done thoroughly and carefully, the main argument for the origin of the physics is hardly novel: it dates back to the mean-field theories by Nevidomskyy et al. which are ten years old and have never been further developed since then.

Based on this as well as the history of research on YbAlB₄, I think that it is very unlikely that this manuscript will influence thinking in the field of the f-orbital materials. This is why I do not recommend it for the publication in Nature Communications.

Answers to the reviewers:

Reviewer #1

First of all, we are pleased that the referee found the topic of the manuscript interesting, and that he/she suggested the improvements which would make our manuscript suitable for Nature Communications. Following this advice, we have made changes to the manuscript, but we will also respond to his/hers questions directly:

The manuscript submitted by Grbić et al. consists of a detailed study of an interesting compound α -YbAlB₄ by several measurements (resistivity, thermal expansion, NMR, magnetization and specific heat).

Quantum criticality has been a topic of discussion and exploration for a long time. In this matter, Yb-based heavy fermion systems are of special interests. Special is this two brother compounds (α -YbAlB₄ and β -YbAlB₄). β -YbAlB₄ has been explored a lot because of its fascinating characteristics. Whereas, this present manuscript deals with the other sibling α -YbAlB₄. The authors have found quantum critical behavior along with a Lifshitz transition under low applied magnetic field and attributed the quantum critical behaviors to the anisotropy of hybridization.

The referee listed out some comments / suggestions / questions which are the followings

1. The referee would suggest to indicate the critical fields (B_1 and B_2) in all the field-dependent plots which will be helpful for the readers.

The referee's request has now been implemented into the figures as two vertical lines marking the magnetic field values of B_1 and B_2 .

2. Why the authors did not show Grüneisen ratio which is a better parameter to see quantum criticality. Such a plot will be necessary to support the claim of the authors.

Unfortunately, the Grüneisen ratio (whether it is taken in its standard form or the “magnetic” form) is not a good variable for this system, because at temperatures below ~200 mK there is a large nuclear contribution to the specific heat (please see Kuga et al., Sci. Adv. **4**, eaao3547 (2018) and Supplemental figure S9, therein). As we increase the magnetic field from 0 to B_2 , the nuclear contribution dominates specific heat at such low temperatures, and because of it the remaining electronic contribution (and the corresponding Grüneisen parameter) have a large uncertainty.

However, this is why we show the thermal expansion coefficient data that changes sign across B_2 , which is one of the prerequisites for the quantum critical signature as it indicates entropy accumulation. Although this is not a sufficient condition, it is a necessary one. With the lack of information from specific heat, we employ a multitude of other techniques to provide evidence that the behaviour at B_2 is indeed related to quantum criticality.

We have now added a sentence explaining this in the manuscript.

3. *The referee would suggest to plot α_c/T as a function of temperature in the place of figure 1(b).*

We have replaced the data shown in the main panel of Fig1b with α_c/T .

4. *Is there any possibility of having Lifshitz transition even at B_2 along with B_1 ? The referee is curious because the evidence of Lifshitz transition was claimed to be seen only from the kink in Fig 3(b), so if there is also a Lifshitz transition at B_2 , then that would be merged with the oscillations. As a result both at B_1 and B_2 a change, though small, has been seen in the $11\nu Q$.*

The most unambiguous evidence of a Lifshitz transition is the appearance of a new QO frequency above B_1 . As the referee noted this is most clearly observed in Fig. 3b and Fig. 3d. On the updated version of the manuscript we have annotated the fields corresponding to B_1 and B_2 . Also, it is visible that at B_1 there is an anomaly in magnetostriction λ_a values in the ab plane (alone), that does not drastically change from 4.2 K and 0.1 K. Such (temperature) dependence is consistent with a Lifshitz transition while its manifestation in only λ_a is consistent with the appearance of a 2D Fermi surface. It is important to note that at B_1 the magnetic field dependence of thermal expansion (in the inset of Fig. 1b) does not show any anomaly with magnetic field, in contrast to the behaviour at B_2 , which also supports the explanation of a Lifshitz transition at B_1 .

At B_2 there is no new oscillation frequency appearing; instead, the oscillations at fields above B_1 agree with what one would typically expect for quantum oscillations, i.e. approximately linear growth toward lower $1/B$ values shown in Fig. 3b. Other transport measurements, such as the Hall effect, do not show clear evidence for a Lifshitz transition at B_2 . The change of Hall effect from B_1 to B_2 shows a large change, but one inconsistent with a Lifshitz transition at B_2 . Magnetostriction (λ_c) shows an anomaly at B_2 , but this time only in c axis, and thermal expansion changes sign at B_2 . Therefore, we concluded that there is unlikely to be a second Lifshitz transition at B_2 .

5. *Page 5, 3rd paragraph: Only from the magnitude of $1/T_1T$ it is not reasonable to say about the closeness to magnetism, especially comparing the magnitude with other compounds where the hyperfine coupling constants are different.*

We have removed discussion on the magnitude of $1/T_1T$. The strongest argument that shows α -YbAlB₄ is not close to magnetism is that there is no change in the NMR spectra that would be caused by the symmetry breaking of magnetic order. Also, the onset of magnetic order would be followed by changes in Knight shift of ¹¹B (¹¹K) or an anomalous change in NMR linewidth, which is not visible at all (these data are now added to the Supplemental). Temperature dependence of ¹¹K shows that below

temperature of ~ 8 K ^{11}K reaches a saturated value for magnetic fields lower than ~ 4.5 T. Such saturation would usually indicate the system enters a Fermi liquid state, but as the $1/T_1T$ diverge at B_1 and B_2 , it indicates that excitations at the Lifshitz transition and quantum critical region are located at $q \neq 0$. Nevertheless, the saturated value of ^{11}K and the behaviour of the NMR linewidth proves there is no static magnetic order emerging.

6. *The referee would suggest include the power-law lines in figure 2(b) of $1/T_1T$.*

We have now introduced power-law lines in figure 2(b).

7. *The temperature dependence of spectra, K (%) and line width are important to show down to low temperatures along with the spectra as a function of the field at the lowest temperature to nullify the possibility of having short-range or long-range ordering at low temperature and across the critical fields. Furthermore, the $1/T_1T$ vs K (%) plot will be helpful to uncover the nature of correlations (FM/AFM). In this respect, the sentence "As seen... B_2 " in 6th Page 2nd paragraph seems also not a proof of the absence of $q=0$ mode as most of the features are rather weak.*

We have now added to the Supplemental data the temperature dependence of ^{11}K for various magnetic fields, the determined full-width-at-half-maximum (FWHM) temperature dependence for the measured ^{11}B site at several magnetic fields and its magnetic field dependence taken at 142 mK. These all show in more detail that there is no anomaly that would correspond to the onset of the magnetic $q=0$ mode. FWHM shows *no* temperature dependence at any magnetic field, while it shows a slowly increasing linear magnetic field dependence up to 4.5 T. This dependence indicates that there is a small static distribution of Knight shift values, but since the absolute value is still very narrow (ranges from ~ 15 kHz to ~ 45 kHz) this most probably originates from a small amount of defects in the crystal and cannot be related to a short-range magnetic order.

Although it is desirable to take whole NMR spectra such as those shown in Fig. S-1 at lower temperatures, this becomes extremely time consuming as we have to reduce rf power and long repetition time to avoid heating of the sample. We believe that the FWHM data taken on the reasonably well separated line of site D are sufficient to claim absence of magnetic order.

From the shown Knight shift data it can now be seen that for all the measured magnetic field values $K(\%)$ saturates to a constant value below 1 K. This means that the $1/T_1T$ vs K (%) plot will not help us determine the nature of the correlations.

8. *In general, as the authors know that the spin-lattice relaxation process along c direction is driven by the spin-fluctuations perpendicular to c -direction, so to have an understanding about the critical fluctuations along different directions, the referee*

would suggest to estimate the spin-fluctuations only along c-axis for which relaxation measurements perpendicular to c is essential. This is of special importance as the system is highly anisotropic as far as the criticality is concerned.

Unfortunately, because of the extremely strong Ising anisotropy of YbAlB_4 , magnetic fields perpendicular to the c direction do not couple to the magnetic moments of f -electrons. This is most clearly demonstrated by the temperature independent magnetic susceptibility for $H \parallel ab$ in the whole temperature range below room temperature (ref. 9). Therefore, the field must be applied along c direction in order to investigate field-induced quantum phenomena. Because of the Ising nature of the system the phase transitions we observe for $H \parallel c$ will not occur for $H \parallel ab$. This is confirmed for example by the 2D nature of the Fermi surface that emerges through the Lifshitz transition at B_1 . As the sample is tilted with respect to the magnetic field, the transition moves to very high fields (this is displayed in the Supplemental figure S-10). In turn, this means that the nature of spin fluctuations will change and that the comparison for $H \parallel c$ and $H \parallel ab$ (or subtraction) will not give any new information.

If the hyperfine coupling were isotropic, the spin fluctuations along the c -axis would never be detected by $1/T_1T$ for $H \parallel c$. However, due to the low local symmetry of the boron sites, an off-diagonal term of the hyperfine coupling tensor is allowed and it couples the critical fluctuations in the c -axis direction to the $1/T_1T$ measured for magnetic field along the c -axis direction. However, at the same time the boron sites are coupled to two Yb sites which are related by mirror symmetry of the ab plane. Therefore, the off-diagonal coupling cannot be determined through Knight shift measurements as the contributions cancel out.

9. Page 5, 2nd paragraph: The exponent of $1/T_1T$ is indeed very small but also close to e.g. YbRh_2Si_2 (0.5) near QCP and the low value of exponent may occur due to other reasons as well, not reasonable to claim that the criticality is nonmagnetic only from that exponent. To claim a close proximity to magnetic order, the unchanged spectra across B_1 and B_2 is important, if so. Though low-temperature spectra are not shown and should be shown. Furthermore, it also mimics the behavior with a slow increase of $1/T_1T$ in case of CeFePO at a metamagnetic transition with a possibility of fermi surface instability (PRL 107, 277002 (2011)). Also see the case for $\text{Sr}_3\text{Ru}_2\text{O}_7$ (PRL 95, 127001 (2005)). Can the authors put light on the possibility of such a scenario in this system? It is more likely that the B_1 and B_2 are metamagnetic transitions similar like the mentioned systems. Again, the referee believes that it is really important to estimate spin-fluctuations only along c -axis to say something concretely about the power law and the divergences of $1/T_1T$ at B_1 and B_2 .

Our statement about an exponent of the temperature dependence of $1/T_1T$ was somewhat imprecise. What we intended to do is to compare our case to the low-dimensional insulator systems, but as it caused confusion, this part is now removed from the manuscript.

In relation to question 7 and this one, we have now added Knight shift data and FWHM data in the Supplemental to show there is no proximity to magnetic order.

The current phenomena cannot be interpreted as metamagnetic transitions (MMT) since several of its aspects are not consistent with it – there is no large anomaly in thermodynamic quantities; the macroscopic susceptibility shows only a small rounding at B_2 and without a drastic increase that is expected at MMT; there is no anomaly registered at B_1 by macroscopic susceptibility measurements. Also, since the effective mass of the quasiparticles is quite low, it is difficult to find a reasonable driving mechanism of the MMT in such a case – from Fig. 6 of the mentioned PRL 107, 277002 (2011), B_1 and B_2 should exceed 20 T.

Regarding the comparison to the systems mentioned by the referee:

1. The QCP of YbRh_2Si_2 occurs at the boundary between the AFM and paramagnetic phases, i.e. accompanied by a symmetry change. This is clearly distinct from the situation in our case of $\alpha\text{-YbAlB}_4$, which does not show any symmetry breaking.
2. Metamagnetism observed in CeFePO and $\text{Sr}_3\text{Ru}_2\text{O}_7$, also shows no symmetry breaking therefore they have some similarities with $\alpha\text{-YbAlB}_4$. However, in $\alpha\text{-YbAlB}_4$, the magnetization changes very smoothly at B_1 and B_2 and does not show any singularity, in sharp contrast to CeFePO and $\text{Sr}_3\text{Ru}_2\text{O}_7$, where the discontinuous jump of M points to a phase transition. The quantum criticality in $\alpha\text{-YbAlB}_4$ shows up only in spin or charge dynamics ($1/T_1T$ or electron-electron scattering rate). These observations place $\alpha\text{-YbAlB}_4$ in a new class of QC materials.

Regarding the estimation of the spin fluctuations – unfortunately this is not possible in $\alpha\text{-YbAlB}_4$, because of the reasons explained in the answer to question 8.

10. From $1/T_1T$ data, it is at all not the case that in the field range between B_1 and B_2 , $1/T_1T$ behaves as a standard metal because $1/T_1T$ at 3 T shows a clear indication of critical fluctuations. This might suggest that B_1 and B_2 might not also be two separated phenomena.

The ratio of the panels in Fig 2b was previously not well chosen as the y-axis was twice as larger than the x-axis, while the ratio of maximum and minimum value of the y-scale is ~ 6 and the ratio of maximum and minimum value of the x-scale is ~ 1000 . This skewed the presentation of the temperature dependence, and we have now corrected it by changing the aspect ratio of the graphs.

We note that also, unfortunately, in the previous version of Fig 2b the measurement at 3 T showed several badly taken points below 100 mK, and by error we did not present all the measured $1/T_1T$ points at 1 T and 1.5 T. This is now corrected.

As can be seen from the figure now, $1/T_1T$ rapidly changes as we approach $T^* \sim 8$ K from high temperatures. For fields away from B_1 and B_2 , below T^* the temperature trend of $1/T_1T$ slows down and from 2 K to 0.1 K it changes from $0.7 \text{ s}^{-1}\text{K}^{-1}$ to only $0.9 \text{ s}^{-1}\text{K}^{-1}$.

This is visible even at 1 T and 1.5 T measurements. So we can safely say that from high to low temperature $1/T_1T$ saturates to a constant value which is consistent with FL. At fields of 1 T and 1.5 T, where the system surely enters a FL ground state, the saturated value is reached below ~ 200 mK, which is consistent with the resistivity data on this compound (Y. Matsumoto et al., Phys. Rev. B **84**, 125126 (2011)). This behaviour is drastically different from the temperature dependence at B_1 and close to B_2 , and this change in behaviour is visible at the upper panel of Fig 2a.

11. With a more closer look at Figure 2(a): why the magnitude of $1/T_1T$ at 142 mK have been plotted, it should be the lowest temperature measured. In that case, the value at 4 T is higher than that of at 3.6 T. How about the stretch exponential in fitting the recovery curves? Can the authors say something about the error in determining T_1 ? The referee would suggest to include some recovery curves at low temperatures in the supplemental materials.

We have chosen to measure the magnetic field dependence of $1/T_1T$ above the base temperature as it was possible to maintain it stable for a long period of time needed for T_1 measurements. These points were taken separately from the temperature dependence measurements, and this enabled us to have a better resolution of the $1/T_1T$ magnetic field dependence. At the base temperature, the dilution refrigerator has the lowest cooling power and as such becomes sensitive to any heating effect, e.g. caused by the eddy currents from varying the magnetic field value. We would be required to wait for the system to stabilize before initiating the T_1 measurement which would prolong the measurement. Also, we would need to measure every T_1 point much longer to avoid heating of the sample with pulses during the T_1 measurements. With the T_1 values we have in this system it would make the measurement drastically longer and susceptible to transient heating effects.

The error bars of determined T_1 values were not shown in Fig 2b, since it would make the graphs less clear. However, we agree that the error bar size should be stated, and this statement has now been added to the figure caption: in the current version of the manuscript they are typically of the size of symbols for data below 1 K, and half that value for the data above 1 K. This is also visible in the upper panel of Fig 2a.

When the data presented in Fig 2b (temperature dependence) was taken in the dilution refrigerator the sample's c axis was slightly misaligned (by $\sim 1^\circ$) with respect to the magnetic field, while the data taken in the upper panel of Fig 2a (magnetic field dependence) was precisely oriented. This is because the sample was mounted on a single-axis rotator for which we could not control the second axis. The misalignment was also noted in the quadrupolar splitting data presented in the Supplemental figure S-5a. Such a small misalignment is experimentally unavoidable and has no influence on the main conclusions of the paper. However, given that the system is an Ising one, the divergence of $(T_1T)^{-1}$ temperature dependence has caused that the data taken at 3.6 T and 4 T, presented in Fig 2b, are of comparable size. Nevertheless, they are within the error

bars (symbol size) of the measured points, and we felt there was no reason to adjust the measured data points.

We have now added in the Supplemental data several recovery curves and a graph showing a stretch exponential coefficient (β). It is visible that the β coefficient reduces as the magnetic field is increased, which is indicating a growing inhomogeneity in the dynamical susceptibility with magnetic field. While at low field the data can be fit by a single component T_1 recovery curve (that consists of three exponentials required by the satellite transition of nuclear spin $I=3/2$), the distribution of T_1 values slowly increases with magnetic field. Given the recent Mössbauer data from reference 36, this broadening of T_1 distribution might originate from the change in Yb electrical quadrupole moment.

12. Can the authors say somethings about the fact that frustration is not at all affecting these critical behaviors? The referee finds it rather important, also to explain the anisotropic nature of criticality, especially in case of thermal expansion coefficient, as have been seen in CeRhSn (Sci. Adv. 1, e1500001 (2015)). The crystal structure α -YbAlB₄ has, if one considers the second nearest neighbor exchange (which seems to be reasonable to consider), then frustration can appear.

In α -YbAlB₄ the Kondo scale, i.e. in the valence fluctuation scale, is ~ 150 K, and at temperatures below there is no localized moment that could cause frustration effects.

When compared, it can be seen that CeRhSn and α -YbAlB₄ have some similarities, but even more differences. Both compounds are Ising type systems, where this strong magnetic anisotropy governs the anisotropy of the overall properties. However, while CeRhSn has a geometrically frustrated lattice (when looking at the nearest neighbours), α -YbAlB₄ does not. For frustration to exist in α -YbAlB₄ it would require a non-negligible next nearest neighbours' interaction, but since α -YbAlB₄ is a non-localized f-electronic system, with conduction electrons acting as interaction mediators, this interaction is not expected to be a significant factor here.

The indication that in CeRhSn magnetic frustration was playing a role was relatively straightforward. Susceptibility shows an Ising-like type of behaviour, together with a large anisotropy in resistivity ($\rho_{ab}/\rho_c \sim 6$). However, the thermal expansion diverges along the a axis, while the one along the c axis stays inert. This indicated that something was happening within the distorted kagome plane. Further analysis and data revealed that frustration was driving the quantum criticality (please see the more detailed discussion our response to question 13).

In α -YbAlB₄ susceptibility shows an Ising type of behaviour, together with an even larger anisotropy in resistivity ($\rho_{ab}/\rho_c \sim 11$). However, the thermal expansion along the c axis is the dominant one (Y. Matsumoto et al., J. Phys.: Conf. Ser. **807**, 022005 (2017)) that shows signature of criticality – hence there is no indication that the 2D hexagonal lattice is somehow involved in the transition at B_2 .

Therefore, in the manuscript from the data of several techniques we conclude that the driving mechanism behind the anisotropy in quantum criticality is the anisotropic (q -dependent) f - c hybridization. This is a different mechanism from CeRhSn, and enables quantum criticality to emerge in an intermediate valence system even without frustration.

13. *The referee also would like to mention that the effect of anisotropy in quantum criticality has been seen also in other systems e.g. CeRhSn and also in case of Fe doped CeRuPO (J. Phys. Soc. Jpn., 82, 033704 (2013)).*

Indeed, these systems were important to our understanding of α -YbAlB₄ (refs. [26] and [37] in the main part of the manuscript), since the criticalities have similar signatures.

What was unexpected about CeRhSn was that the transport anisotropy ($\rho_{ab}/\rho_c \sim 6$) is larger in CeRhSn than in CeRhIn (M. S. Kim et al., Phys. Rev. B **68**, 054416 (2003) and H. Higaki et al., J. Phys. Soc. Jpn. **75**, 024709 (2006)), while at the same time the hybridization strength is lower in CeRhSn than in CeRhIn (K. Shimada et al., Phys. B **378-380**, 791 (2006)). This indicated that the anisotropy in resistivity is not caused by hybridization strength.

Also, from thermal expansion studies (Y. Tokiwa et al., Sci. Adv. **1**, e1500001 (2015) and R. K uchler et al., Phys. Rev. B, **96**, 241110 (2017)), it is visible that by applying a pressure within the kagome plane (i.e. ab plane) anisotropy is decreased. It was concluded that this is caused by the release of frustration, in contrast to the pressure applied along the c axis which did not qualitatively affect the system. The low temperature thermal expansion shows a divergence within the kagome plane even though the system is close to an Ising one along the c axis – this is another indication of the importance of the 2D lattice in this system.

For the all these data, it was concluded that the anisotropy in CeRhSn originates from 2D magnetism and not from f - c hybridization.

Regarding CeRuPO and CeFePO, they are both Kondo lattice systems, with a rather complex magnetic phase diagram (that evolves with doping) due to the interplay of crystalline electric field level and the magnetic RKKY exchange interaction.

What makes α -YbAlB₄ special is that it is an intermediate valence (IV) system (unlike CeRuPO) without a clear frustration (like CeRhSn) that could enable quantum criticality. Therefore, it is *a priori* not expected that we could tune it easily with low magnetic field. However, the observed Lifshitz transition (which has not been seen before in an IV system at such a low magnetic field) and the survival of coherent quantum critical fluctuations, without symmetry breaking, indicates that a new physical phenomenon has emerged in this system.

If α -YbAlB₄ was a well renormalized heavy fermion compound with higher effective mass, our finding would not be so impressive. But, the specific anisotropy of the measured quantities in this compound (e.g. $\rho_{ab}/\rho_c \sim 11$), together with additional information from HAXPES (Kuga et al., PRL **123**, 036404 (2019)) and ARPES, indicates the presence of a k-dependent anisotropy of the hybridization potential $V_{\alpha\alpha}(\mathbf{k})$.

Reviewer #2 (Remarks to the Author):

The manuscript “Anisotropy-driven quantum criticality in an intermediate valence system” by Grbic et al presents thermodynamic, magnetotransport and microscopic experiments on α -YbAlB₄ in an applied magnetic field (B). The authors observe two features at $B_{c1}=2.1$ T and $B_{c2} = 3.6$ T and associate them with a Lifshitz transition and a quantum critical point, respectively. The authors then attribute the quantum critical behavior to the anisotropic hybridization between f- and conduction electrons. Efforts into determining the origin of quantum critical behavior in strongly correlated systems are important, and the breadth of low-temperature techniques reported in this manuscript is remarkable. There are, however, important questions that need to be answered to demonstrate the authors’ claims and to meet Nature Communications’ criteria:

1) A central point of the manuscript is that the origin of quantum criticality is the anisotropic hybridization between f- and conduction electrons. I do agree that anisotropic hybridization is a key parameter in f-based systems because it is a property of the ground state wavefunction which inevitably determines Kondo and RKKY interactions. Anisotropic hybridization per se, however, cannot generate quantum criticality, but it rather allows for the fluctuations that drive criticality. The fundamental parameter to characterize quantum criticality is a characteristic length scale which diverges at the quantum critical point. What is this length scale in α -YbAlB₄?

We agree with the Referee that the anisotropy of the hybridization does not, by itself, guarantee quantum critical behaviour. We would like to emphasize, as we do in the manuscript, that the role of the anisotropy here is to enable the emergence of the quantitative behaviour of various observables in the vicinity of the critical point, i.e. the critical exponents. In particular, as it was shown by HAXPES (Kuga et al., PRL **123**, 036404 (2019)), and as is explained in the manuscript, the anisotropy of the hybridization is similar in α - and β -polymorphs of YbAlB₄, and therefore one might expect the critical exponents to be the same in the two compounds. However, since the many-body ground state in α - and β -YbAlB₄ is different, the values of the exponents of the temperature-dependent resistivity are somewhat similar ($n=1.65$ in α -YbAlB₄ at B_2 , and $n=1.5$ in β -YbAlB₄), but they are for different directions. In α -YbAlB₄ $n=1.65$

is for resistivity in c axis (while the resistivity in ab plane shows FL behaviour with $n \approx 2$), and in β -YbAlB₄ exponent of $n=1.5$ is for resistivity in the ab plane.

To answer the Referee's question: the characteristic length scale would be the correlation length of the magnetic fluctuations which is expected to diverge as the magnetic field strength approaches $B_2 \sim 3.6$ T. However, since there is no symmetry breaking, and the data show no short-range order forming, there is no visible length scale that can be associated to the divergence at B_2 . Therefore, the field-tuned quantum criticality is of novel type, and observed in a temporal scale, i.e. the dynamic critical exponent.

2) Anisotropic hybridization also is relevant for the formation of a heavy-fermion state and, if the state at about Bc_2 has Kondo correlations, then one would expect enhanced effective masses. The authors, however, obtain a light effective mass ($m^ = 0.55 m_e$). This is a contradiction that the authors need to address.*

The quantum oscillations we measure at low fields around B_1 can only pick up the contribution to the Fermi surface of the carriers with lightest mass. This is natural considering that the amplitude of the oscillations, as given by the Lifshitz-Kosevich formula, goes proportional to

$$A(T) \sim \frac{x(T)}{\sinh(x(T))} \sim T \exp\left(-\alpha \cdot m^* \cdot \frac{T}{B}\right),$$

where the dimensionless parameter $x(T) = \alpha \cdot m^* \cdot T/B$ (with $\alpha = 14.69$ T/K determined by universal constants). Therefore, the amplitude of oscillations is exponentially suppressed with the increasing effective mass. This is well known in heavy fermion systems, where one must go to large fields and ultra-low temperature in order to measure m^* of the order of 10 and above.

So the low value of m^* corresponding to the Fermi surface pocket appearing above $B > B_1$ is not a contradiction, and it does not mean that there are no heavier Fermi surface sheets — we know for a fact that those exist. However, it is not the heavy Fermi sheets that drive the Lifshitz transition at B_1 . Rather, our analysis points to the Lifshitz transition being driven by the Zeeman splitting of a tiny Fermi surface pocket whose total area, corresponding to the oscillation frequency $F \sim 10$ T, is a tiny portion (less than 1%) of the total Fermi surface volume. The fact that this pocket is tiny is why the effects of the Lifshitz transition are so subtle, with only a small signal observed in resistivity (see Fig. 2c). Nevertheless, the Lifshitz transition has a clear signature in NMR relaxation rate $1/T_1$ (see Fig. 2a, 2b), which is what allowed us, together with quantum oscillations, to unambiguously determine the nature of this signature at $B=B_1$.

3) Related to the second point is the theoretical remark that the f electron appears to become detached from the Fermi energy above 3 T in α -YbAlB₄ (Ref. 12 of the manuscript). In this case, quantum critical fluctuations would not be expected

We believe that there is a misunderstanding originating from an imprecise statement in our manuscript (this has now been corrected).

The extension of the physical model described in Ref. 12 (now ref. 13) is connected to the Lifshitz transition at $B_1 = 2.1$ T in α -YbAlB₄, as a remnant form of the criticality in β -YbAlB₄. This was described on pages 3 and 9. In this case, the language is imprecise — it's not that the f -electron becomes "detached from the Fermi energy", rather the statement ought to be that in the field of B_1 the Zeeman energy becomes resonant with the position of the f -level relative to the Fermi energy. However, make no mistake — it's not that in the field above B_1 the f -electrons cease playing a role, on the contrary, their contribution to transport and other quantities becomes most pronounced at that field. What one ought to remember is that the f - c hybridization strength is large (here of the order of 20 meV), so in order to "detach" the f -electron from the Fermi energy properties, one would need to apply a huge magnetic field, such that its Zeeman energy would become comparable to the hybridization strength. This is clearly an unphysically large-field regime and it is not what has been observed in the experiments on either α or β -YbAlB₄.

That said, in α -YbAlB₄ at B_1 we observe only the Lifshitz transition, without the quantum criticality, but not because f -electron was detached from the Fermi energy.

4) The authors state that minor changes in valence are observed to 40 T, but they fail to mention these x-ray absorption measurements were performed at 2K, which is above the region in which the effects reported here are pronounced. This is misleading and should be clarified in the manuscript.

To put our statement about the X-ray absorption measurements in context, it was said in relation to observing an anomaly in the ²⁷Al and ¹¹B quadrupolar coupling (v_Q) of the NMR signal in the vicinity of B_2 (figures shown in figures S-5 of the Supplemental). If the observed anomaly is because of the valence transition, then it should be observed by XAS since they are also observed at 4 K. Although these data are not shown explicitly in the paper, the Supplemental figure S-3 shows two sets of points: yellow points are measured only at 142 mK, while purple points show an average value taken at each magnetic field for all the temperatures measured below 4 K (more than 20 measured values) to reduce the data scatter. The averaging was allowed to be done because the anomaly in v_Q is visible as high as 10 K, but below 4 K the temperature dependence saturates to a constant value. Therefore, if the anomaly in v_Q corresponds to a magnetic field-induced first order valence change in this system it should be detected by XAS also at 2 K.

If the question of the Referee is not related to the reported anomaly in the ²⁷Al and ¹¹B quadrupolar coupling, then the first order transition that was not detected by XAS would have been seen by quadrupolar coupling of ²⁷Al or ¹¹B, or both.

We believe this answers the Referee's question, but we would also like to say that in the hypothetical case that at B_2 α -YbAlB₄ is located in the vicinity (below 2 K) of the valence quantum critical point, it is expected to see a broad magnetic field-induced response detectable at relatively high temperatures. For instance, in a study on

YbAgCu₄ (Nakamura et al., J. Phys. Soc. Jpn. **81**, 114702 (2012)), XAS observed a finite broad magnetic field-induced change in valence already at 50-55 K (as seen in Fig. 9 of the paper), even though the valence change onsets at $T_V \sim 42$ K. This has been similarly observed in the YbInCu₄ system.

Another excellent example is the HAXPES measurement of a first order transition induced by Fe doping in α -YbAlB₄, (Kuga et al., Sci. Adv. **4**, eaao3547 (2018), Ref. 22 in the manuscript and Ref. 5 of the Supplemental), where the transition was detectable at very high temperature (even at 20 K), even though the phenomenon is at $T=0$ K. XAS is more bulk sensitive than HAXPES when the measurement is done in conventional transmission geometry as it was.

5) No strong evidence for a Lifshitz transition is presented. The authors do seem to appreciate this issue initially stating that their results indicate a Lifshitz transition, but then a much stronger statement is made: “The discovery of a Lifshitz transition in an intermediate valence system at such low field alone is remarkable (...)”. This is also misleading. The authors should soften their statement as a change in the density states in field does not guarantee a Lifshitz transition.

The evidence of a Lifshitz transition is shown in our presentation of quantum oscillations (Figs 3a and 3b) which demonstrate the appearance of a new frequency with a strikingly quasi-2D nature above B_1 . We are not aware of a more direct probe of magnetic field induced changes in the Fermi surface. The referee is correct to note that the changes in the density of states is small, and that by itself this is insufficient to claim a Lifshitz transition. However, from the measured effective mass (Fig. 3c) and geometry (Fig. 3d) we showed that the changes in density of states and magnetization are in quantitative agreement with quantum oscillations. Therefore, we concluded that the evidence for a Lifshitz transition is unambiguous. When these observations are put together with the measured behaviour of magnetostriction (Fig. 2d), thermal expansion (Fig. 1b) and NMR (Fig. 2b), it is shown that this conclusion is consistent and valid. To address the referee’s concern, we have added a sentence: “Therefore, considering both the primary evidence from QOs together with the consistency with macroscopic measurements, the results overwhelmingly indicate a Zeeman driven Lifshitz transition occurs at B_1 .”

6) Another important aspect of anisotropic hybridization is the actual shape of the ground state doublet wavefunction. The authors, however, do not discuss in detail the groundstate wavefunction of α -YbAlB₄ and its comparison to β -YbAlB₄. These two systems are locally isostructural but have distinct groundstates (e.g. Ref 14 of the manuscript).

Ref. 14 (now Ref. 15) does not make any claims that the nature of the ground state Yb doublet in α -YbAlB₄ is any different from that in β -YbAlB₄. In fact, the ab initio calculations and the symmetry analysis suggest that the ground state doublet should be the same in both compounds; primarily $|J_z = \pm 5/2\rangle$ of the total angular momentum

$J=7/2$ on Yb. The nature of ground state doublet and the crystal field splitting is determined by local chemistry and point-group symmetry around the Yb site, and as the Referee remarked, those are locally isostructural in the two compounds.

However, the many-body ground state is different in α - and β -YbAlB₄. While both compounds have strong f - c hybridization, the position of the renormalized f -doublet relative to the Fermi level, ϵ_f^* is different, as remarked earlier in response to point (1) of the referee. As a result, β -YbAlB₄ is a critical metal, with $\epsilon_f^* = 0$, whereas α -YbAlB₄ appears to be a canonical heavy-fermion Fermi liquid *in zero field* (stable up to at least ~ 1.5 T). This all changes upon the application of the Zeeman field corresponding to B_2 — at that point, α -YbAlB₄ becomes quantum critical, which is the main finding of this manuscript.

Regarding the different many-body ground states and its connection to the shape to the doublet wavefunction, we discuss it indirectly in the last part of the manuscript, on pg. 10. There we consider the universal presence of (various forms of) quantum critical signature in β -YbAlB₄ in different conditions (under ambient conditions, with doping and under increased pressure) and in α -YbAlB₄ also in different conditions (without doping but under magnetic field, and with doping). Since quantum criticality is typically not easy to find in intermediate valence compounds (please also have a look at our answer in question 8), and in the YbAlB₄ family it is in principle “difficult to avoid”, it naturally implies the presence of a robust energy scale insensitive to small variations of local chemistry that is essential for quantum criticality to emerge. The only energy scale of the system that matches this description, and can explain the presented data, is the anisotropic hybridization. HAXPES data of Ref. 14 (now Ref. 15) have shown that there is only a small difference of the ground state wavefunction (or more specifically, its admixtures) between α - and β -YbAlB₄, which defines the large-energy scale, and therefore the origin of the different many-body ground states is caused by the differences in the local chemistry of the compounds (the small subkelvin scale).

7) Further, the crystal field splitting between ground state and first excited crystal field doublet could also shed light on the evolution of the groundstate wavefunction in magnetic fields because Zeeman splitting may mix different doublets. How are the wavefunctions expected to change in field?

The crystal field splitting to the first excited doublet is large (with the energy gap of $\Delta \sim 80$ K) in comparison to the scale of the ultra-low temperature measurements reported in this manuscript. This crystal field splitting, visible for instance in the Schottky anomaly in the specific heat, as well as in the recent ARPES measurements (in Ref. 14 of the manuscript), is about the same in the α - and β - polymorphs of YbAlB₄. In both cases, the role of the excited crystal field doublet can be safely ignored at temperatures of the order of 1 K.

8) The authors state that the presence of f -electron valence fluctuations to a non-magnetic state is regarded (as) an anathema to quantum criticality. I tend to disagree

with such strong statement as this is not necessarily the case. For instance, CeCu₂Si₂ and PuCoGa₅ have been argued to display superconductivity around a valence quantum critical point.

The statement that valence fluctuations are regarded as an anathema to quantum criticality is used in the introduction of the manuscript to point out that out of many intermediate valence (IV) compounds, accessible quantum criticality is *rarely* displayed. This statement is later (in the introduction) extended to explain how in IV compounds one needs to apply very high pressures or magnetic field to reach quantum criticality, which makes such cases *difficult* to study. Therefore, we express the sentiment of the research community since, for the reasons stated above, IV compounds have been less interesting to study than the heavy fermion compounds.

In CeCu₂Si₂ superconductivity is enhanced by valence fluctuations, but it is not really a representative for the IV compounds. CeCu₂Si₂ is in fact a heavy fermion compound at ambient pressure (with a Kondo temperature of ~20 K), and the valence quantum critical point is reached with an application of 4.5 GPa of pressure where the compound becomes intermediate valence one (with a valence fluctuating scale of ~150 K). This pressure is not easily achieved (more difficult than to apply a magnetic field of 3.6 T), and a special type of cell was built for its study by transport techniques.

Regarding the case of PuCoGa₅; one way to define IV is by the magnitude of the Kondo temperature, so the definition of what different authors mean by "intermediate valence" has to be taken in the proper context. For instance, in the case of PuCoGa₅ raised by the Referee, the Kondo temperature is actually very low – estimated to be much lower than 10 K. This is why PuCoGa₅ does not have a hallmark Fermi liquid-like susceptibility, but it instead continues to follow the Curie-Weiss law down to 18.5 K when superconductivity sets in. From this perspective, we would respectfully not call PuCoGa₅ an "intermediate valence compound".

To make the answer clearer there are two different names for different valence phenomena (A. C. Hewson, *The Kondo Problem to Heavy Fermions*): intermediate valence (when the valence fluctuates), and mixed valence (when there is a spatial distribution of valence, like in SmS). This distinction was forgotten with time as the focus of the community shifted to other topics, but in the manuscript, we adhere to this interpretation.

Generally speaking, the authors would stand by the statement that in proper IV compounds (defined as those where the Kondo temperature is very large, for instance in YbAl₃), quantum criticality is *typically* not observed (or very difficult to), because of the large Kondo scale involved. What makes YbAlB₄ (both α - and β - polymorphs) different, is that the hybridization function is highly anisotropic and believed to have nodes, i.e. vanishes completely at certain points in the Brillouin zone. Thus, even though the average scale of the hybridization may be large (hence, the adjective "intermediate valent"), quantum criticality can still appear because hybridization vanishes somewhere near the Fermi surface.

Reviewer #3 (Remarks to the Author):

This manuscript reports on experimental studies of the mixed-valence material YbAlB₄. The authors have used quite an impressive range of experimental techniques to perform thermodynamic measurement and by analyzing their experimental data they argue that the alpha-polymorph of YbAlB₄ exhibits quantum critical behavior and Lifshitz transition under an application of low magnetic fields. The authors argue that alpha-YbAlB₄ is unique in that it provides the first example of a system in which quantum criticality co-exists with the mixed-valence. Furthermore, the authors argue that this co-existence is governed by the momentum anisotropy of the hybridization matrix element between the f- and d-orbitals of the Yb ions.

First of all, I would like to note that the several authors of the current manuscript (K. Kuga, S. Nakatsuji, A. Nevidomskyy to name a few) have published their research on this and related materials before. In fact, the first publications date back to almost 10 years ago. Despite that history, the topical has never emerged as a 'main stream' topic, unlike the one of topological Kondo insulators (which is mentioned in the discussion section of the manuscript). From this, I tend to think that despite the current efforts, the earlier and present experimental findings while remaining interesting for the experts in the field of f-orbital based ("heavy-fermion") materials, will unlikely bring this topic into the main-stream of the contemporary condensed matter physics.

While the experimental analysis is broad and appears to be done thoroughly and carefully, the main argument for the origin of the physics is hardly novel: it dates back to the mean-field theories by Nevidomskyy et al. which are ten years old and have never been further developed since then.

Based on this as well as the history of research on YbAlB₄, I think that it is very unlikely that this manuscript will influence thinking in the field of the f-orbital materials. This is why I do not recommend it for the publication in Nature Communications.

The α - and β -YbAlB₄ polymorphs have indeed provided much more novel physical phenomena than first anticipated, and surely more than an average intermediate valence (IV) system would. This is because they are not typical IV compounds and several phenomena are regarded as “first time observed” in these compounds. Each of these phenomena were thoroughly studied exactly *because* they were “first” and *because* the YbAlB₄ family is not “main-stream”. For example:

- First Yb-based HF superconductor: S. Nakatsuji et al., Nat. Phys. **4**, 603 (2008),
- First quantum criticality in ambient conditions Y. Matsumoto et al., Science **331**, 316 (2011),
- Extended “strange metal” phase: T. Tomita et al., Science **349**, 506 (2015),
- A quantum critical first order valence transition: K. Kuga et al., Sci. Adv. **4**, 3547 (2018),

- Experimental verification of an anisotropic hybridization: K. Kuga et al., Phys. Rev. Lett. **123**, 036404 (2019), and others.

Needless to say, work on such a special system had to be done thoroughly to unambiguously determine the mechanism behind the novel phenomena that emerged in the class of intermediate valence systems, which is why their study has been lasting for over a decade.

Although the model of an anisotropic hybridization dates back to 2009, the goal of the current manuscript is to experimentally prove its presence and immense role in α -YbAlB₄ compound. As quantum criticality has, in various forms, been found in β -YbAlB₄ (in pristine samples; under ambient conditions and under increased pressure) and α -YbAlB₄ (in pristine samples, under magnetic field (this manuscript), and if Fe-doped), it can be regarded as universally present in this family. The β -YbAlB₄ has attracted considerable attention since 2008, which was one of the main topics of HF physics. α -YbAlB₄ has been paid much less attention on than the β -YbAlB₄. This is understandable considering the results reported so far. However, our discovery of the Lifshitz transition and the novel QC behaviour should change this situation.

Typically, quantum criticality is not in fact easy to find in intermediate valence compounds, it naturally implies the presence of a very robust energy scale insensitive to small variations of local chemistry that is essential for quantum criticality to emerge. The only such energy scale of the system that matches this description, and can explain the presented data, is the anisotropic hybridization. HAXPES data of Ref. 15 have shown that there is only a small difference of the ground state doublet wavefunction (or more specifically – it's admixtures) between α - and β -YbAlB₄, which defines the large-energy scale, and therefore the origin of the different many-body ground states is caused by the differences in the local chemistry of the compounds (the small subkelvin scale).

In addition, the Lifshitz transition and signature of quantum criticality presented in this manuscript is a novel because it shows no symmetry breaking, and as such cannot be directly related to a diverging spatial quantity. Nevertheless, the divergence in $(T_1T)^{-1}$ indicate that there is a diverging temporal scale, and it is quite non-conventional that coherent fluctuations would survive the many orders of magnitude larger valence fluctuations.

Therefore, we disagree that the current manuscript will not bring about anything new or change the view of thinking in the field.

REVIEWERS' COMMENTS

Reviewer #1 (Remarks to the Author):

After going through the updated draft, I came to the conclusion that the present updated version of the manuscript can be accepted in Nature Communication. They have replied and also took necessary action satisfactorily to modify the manuscript based on my suggestions and questions.

Reviewer #2 (Remarks to the Author):

The manuscript “Anisotropy-driven quantum criticality in an intermediate valence system” by Grbic et al presents thermodynamic, magnetotransport and microscopic experiments on α -YbAlB₄ in an applied magnetic field (B). The authors have done a very good job answering the questions from all referees and strengthening their claim that intermediate valence α -YbAlB₄ likely hosts a new class of quantum criticality driven by anisotropic hybridization.

I emphasize that efforts into determining the origin of quantum critical behavior in strongly correlated systems are important, and the breadth of low-temperature experimental techniques reported in this manuscript is remarkable. I now recommend publication of this article in Nature Communications.